# Apelin signaling drives vascular endothelial cells toward a pro-angiogenic state

Christian SM Helker[1,2]*, Jean Eberlein[1,2], Kerstin Wilhelm[3], Toshiya Sugino[3], Julian Malchow[2], Annika Schuermann[4], Stefan Baumeister[2], Hyouk-Bum Kwon[1], Hans-Martin Maischein[1], Michael Potente[3,5], Wiebke Herzog[4,6], Didier YR Stainier[1,5]*

[1]Department of Developmental Genetics, Max Planck Institute for Heart and Lung Research, Bad Nauheim, Germany; [2]Philipps-University Marburg, Faculty of Biology, Cell Signaling and Dynamics, Marburg, Germany; [3]Angiogenesis and Metabolism Laboratory, Max Planck Institute for Heart and Lung Research, Bad Nauheim, Germany; [4]University of Muenster, Muenster, Germany; [5]DZHK (German Center for Cardiovascular Research), partner site Frankfurt Rhine-Main, Berlin, Germany; [6]Max Planck Institute for Molecular Biomedicine, Muenster, Germany

**Abstract** To form new blood vessels (angiogenesis), endothelial cells (ECs) must be activated and acquire highly migratory and proliferative phenotypes. However, the molecular mechanisms that govern these processes are incompletely understood. Here, we show that Apelin signaling functions to drive ECs into such an angiogenic state. Zebrafish lacking Apelin signaling exhibit defects in endothelial tip cell morphology and sprouting. Using transplantation experiments, we find that in mosaic vessels, wild-type ECs leave the dorsal aorta (DA) and form new vessels while neighboring ECs defective in Apelin signaling remain in the DA. Mechanistically, Apelin signaling enhances glycolytic activity in ECs at least in part by increasing levels of the growth-promoting transcription factor c-Myc. Moreover, *APELIN* expression is regulated by Notch signaling in human ECs, and its function is required for the hypersprouting phenotype in Delta-like 4 (Dll4) knockdown zebrafish embryos. These data provide new insights into fundamental principles of blood vessel formation and Apelin signaling, enabling a better understanding of vascular growth in health and disease.

*For correspondence:
christian.helker@biologie.uni-marburg.de (CSMH);
Didier.Stainier@mpi-bn.mpg.de (DYRS)

## Introduction

Endothelial cell sprouting is a fundamental process of physiological and pathological blood vessel growth. Attracted by growth factors such as vascular endothelial growth factor-A (VEGF-A) secreted from hypoxic tissues, endothelial cells (ECs) break out of the quiescent vessel wall to form new vessel branches (*Ferrara et al., 2003*; *Koch and Claesson-Welsh, 2012*). ECs with higher levels of VEGF-A signaling become invasive tip cells that lead new vascular sprouts, while neighboring ECs with lower VEGF-A signaling become trailing stalk cells (*Gerhardt et al., 2003*). This process is coordinated by Delta-like 4 (DLL4)/Notch signaling. Activation of Notch receptors by their ligand DLL4, expressed by tip cells, represses tip cell behavior in stalk cells (*Hellström et al., 2007*; *Leslie et al., 2007*; *Siekmann and Lawson, 2007*; *Suchting et al., 2007*). Loss of Notch signaling, on the other hand, causes excessive tip cell formation and vascular overgrowth (*Hellström et al., 2007*; *Leslie et al., 2007*; *Siekmann and Lawson, 2007*; *Suchting et al., 2007*).

Apelin (Apln) is a small secreted peptide, which was initially identified because of its inotropic activity (*Szokodi et al., 2002*). *Apelin* was subsequently described as a tip cell-enriched gene

(*del Toro et al., 2010*). Apelin (*Tatemoto et al., 1998*), as well as the newly identified ligand Apela (*Chng et al., 2013*; *Pauli et al., 2014*), can both activate the Apelin receptor (Aplnr), a 7-transmembrane G-protein-coupled receptor (GPCR). Mouse and frog embryos lacking Apln or Aplnr function exhibit reduced vascular outgrowth, decreased EC proliferation, smaller vessel diameter as well as defects in the alignment of arteries and veins (*Cox et al., 2006*; *Kälin et al., 2007*; *Kidoya et al., 2008*; *del Toro et al., 2010*; *Kidoya et al., 2010*; *Kidoya et al., 2015*; *Papangeli et al., 2016*). In addition, Apelin signaling has been implicated in several cardiovascular diseases including pulmonary hypertension (*Goetze et al., 2006*; *Alastalo et al., 2011*; *Chandra et al., 2011*), atherosclerosis (*Hashimoto et al., 2007*; *Chun et al., 2008*; *Kojima et al., 2010*; *Pitkin et al., 2010*), myocardial infarction (*Tempel et al., 2012*; *Wang et al., 2013*; *Zhang et al., 2016*; *Chen et al., 2017*), and tumor angiogenesis (*Kidoya et al., 2012*; *Zhao et al., 2018*; *Uribesalgo et al., 2019*). However, the cellular mechanisms by which Apelin signaling functions within the vasculature remain elusive. Using zebrafish mutants combined with mosaic analyses, high-resolution time-lapse imaging, and metabolic studies, we find that Apelin signaling is required to boost endothelial metabolic activity during angiogenic sprouting. Furthermore, we show that Apelin signaling acts downstream of Notch signaling, where it is required for Notch-controlled angiogenesis.

## Results

### Apelin signaling is required for angiogenic sprouting

To examine the expression pattern of the *apelin* ligand and receptor genes during angiogenic sprouting in zebrafish embryos, we first performed whole-mount in situ hybridization during intersegmental vessel (ISV) formation. We detected clear *alpn*, but no *apela*, expression within the sprouting ISVs (*Figure 1—figure supplement 1* arrowheads). For the receptor genes, we could only detect *aplnrb* expression in the ISVs (*Figure 1—figure supplement 1* arrowheads).

To visualize *apln* and *aplnrb* expression at single cell resolution, we developed reporters using Bacterial artificial chromosome (BAC) recombineering (*Figure 1—figure supplement 2*). To this end, we replaced the ATG of an *apln* containing BAC with an EGFP cassette. Similarly, we replaced the stop codon of an *aplnrb* containing BAC with a tandem fluorescent timer (TagRFP-sfGFP) cassette leading to a fusion protein. We injected both modified BACs into one-cell stage zebrafish embryos to generate stable transgenic lines, *Tg(apln:EGFP)* and *Tg(aplnrb:aplnrb-TagRFP-sfGFP)* (hereafter referred to *Tg(aplnrb:aplnrb-sfGFP)*) (*Figure 1—figure supplement 2*). We first detected weak *apln*:EGFP expression in sprouting ISVs at 30 hpf (*Figure 1A*). At 54 hpf, all ECs within the dorsal longitudinal anastomotic vessel (DLAV) – a vessel formed by tip cells – were labeled (*Figure 1A*, arrowheads) while some stalk cells also exhibited weak *apln*:GFP expression (*Figure 1A*, arrows). Of note, *aplnrb*:Aplrnb-sfGFP expression at 26 hpf was visible in the entire ISV sprout (*Figure 1B* arrowheads), but it was absent from non-angiogenic ECs within the dorsal aorta (DA). At 54 hpf, *aplnrb*:Aplrnb-sfGFP expression was detected in all ECs that had sprouted out of the DA but also weakly in ECs within the DA (*Figure 1B*). These results suggest that *apln* is expressed in tip cells while *aplnrb* is expressed in all sprouting ECs.

To examine the function of Apelin signaling during sprouting angiogenesis in zebrafish, we used mutants for *aplnra* (*Helker et al., 2015*), *aplnrb* (*Helker et al., 2015*), *apln* (*Helker et al., 2015*) and *apela* (*Chng et al., 2013*). Homozygous *aplnra* mutant embryos exhibited no obvious defects during ISV formation (*Figure 1—figure supplement 3*). However, homozygous *aplnrb* mutant embryos exhibited reduced ISV length and failed to form the DLAV (*Figure 1—figure supplement 3*). This phenotype was more severe in embryos lacking both *aplnra* and *aplnrb* (*Figure 1C*, *Figure 1—figure supplement 3*), indicating partial compensation. We also analyzed *apln* and *apela* mutants. Homozygous *apela* mutant embryos displayed only a mild delay in ISV outgrowth (*Figure 1—figure supplement 3*), while homozygous *apln* mutant embryos exhibited defects in ISV outgrowth and failure to form the DLAV (*Figure 1—figure supplement 3*). Loss of both ligands increased the severity of the phenotype leading to ISV stalling at the horizontal myoseptum (*Figure 1C*, *Figure 1—figure supplement 3*). Consistent with studies in the mouse retina (*del Toro et al., 2010*), our studies identify *apln* expression as a marker of endothelial tip cells in zebrafish and show that Apelin signaling is required for angiogenic sprouting.

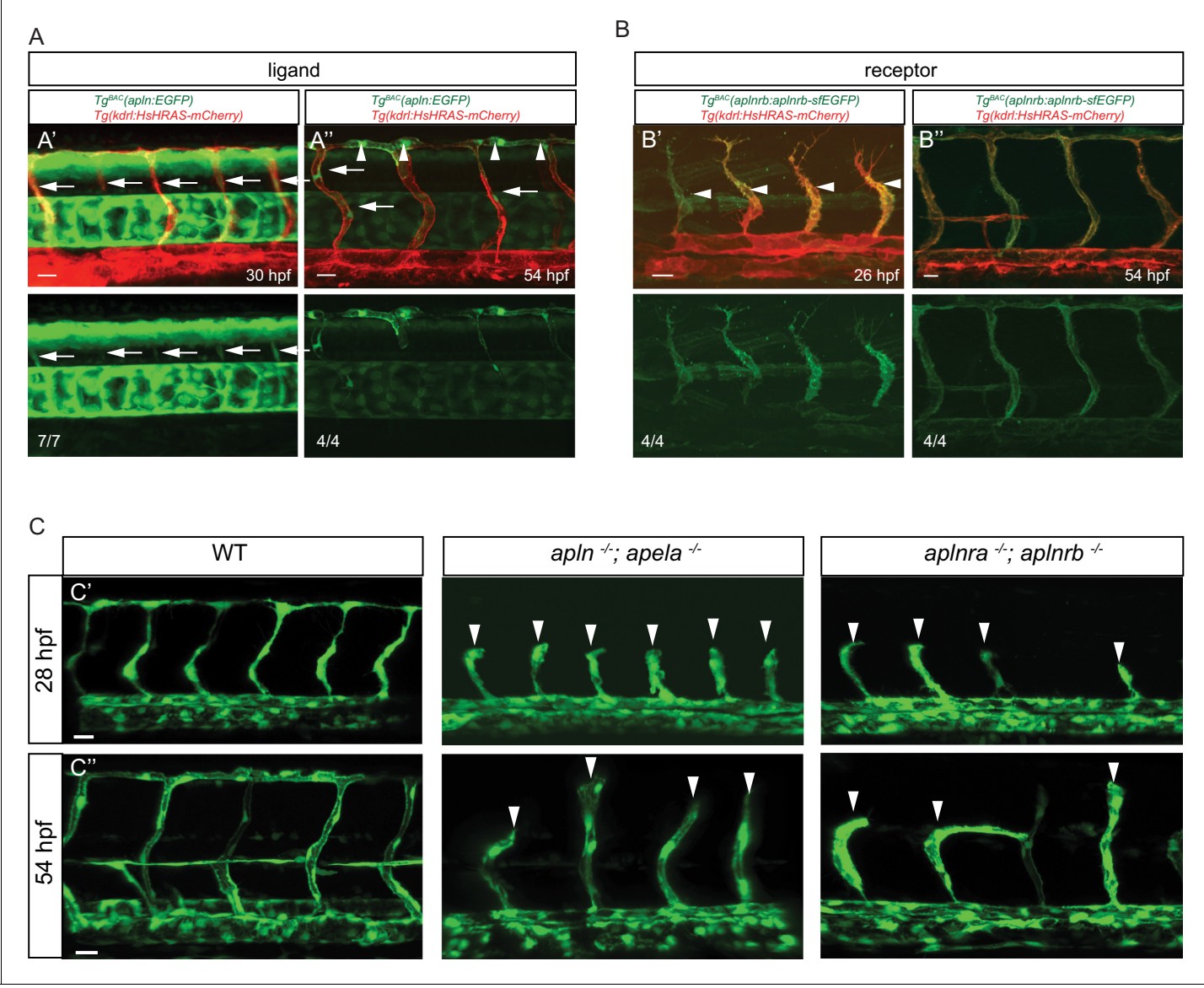

**Figure 1.** Apelin signaling promotes endothelial sprouting. Visualization of *apelin* and *apelin receptor b* expression using transgenic reporter lines. Confocal projection images of the trunk region of zebrafish embryos. (**A**) $Tg^{BAC}$(*apln*:EGFP) expression is detectable in growing ISVs at 30 and 54 hpf. Arrowheads point to strong *apln* expression in tip cells, while arrows point to weak *apln* expression in stalk cells. (**B**) $Tg^{BAC}$(*aplnrb*:aplnrb-EGFP) expression is detectable in sprouting ECs (arrowheads) at 26 hpf and is clearly present in the ISVs and DLAV at 54 hpf. (**C**) Inactivation of Apelin ligand and receptor genes impairs angiogenesis. Confocal projection images of the blood vasculature in the trunk region of *Tg(fli1a:EGFP)* embryos. *apln* $^{-/-}$; *apela* $^{-/-}$ as well as *aplnra* $^{-/-}$; *aplnrb* $^{-/-}$ embryos exhibit a reduction in vascular sprouting at 28 and 54 hpf. Arrowheads point to stalled ISVs. Scale bars: A', 10 μm; A'', B', C', C'', 20 μm; B'', 15 μm.

The online version of this article includes the following figure supplement(s) for figure 1:

**Figure supplement 1.** Expression of apelin ligand and receptor genes by in situ hybridization.

**Figure supplement 2.** Generation of the $Tg^{BAC}$(*apln:EGFP) and Tg(aplnrb:aplnrb-TagRFP-sfGFP)* reporter lines.

**Figure supplement 3.** Quantification of angiogenic defects.

## Apelin signaling regulates tip cell morphology

To investigate when the sprouting defects in Apelin signaling-deficient embryos first appear, we analyzed developmental time points when tip cells start to sprout out of the DA. However, no differences in sprout initiation or tip cell specification were observed in double homozygous receptor or ligand mutants (*Figure 2—figure supplement 1*, *Figure 2A*). Instead, we found that sprout

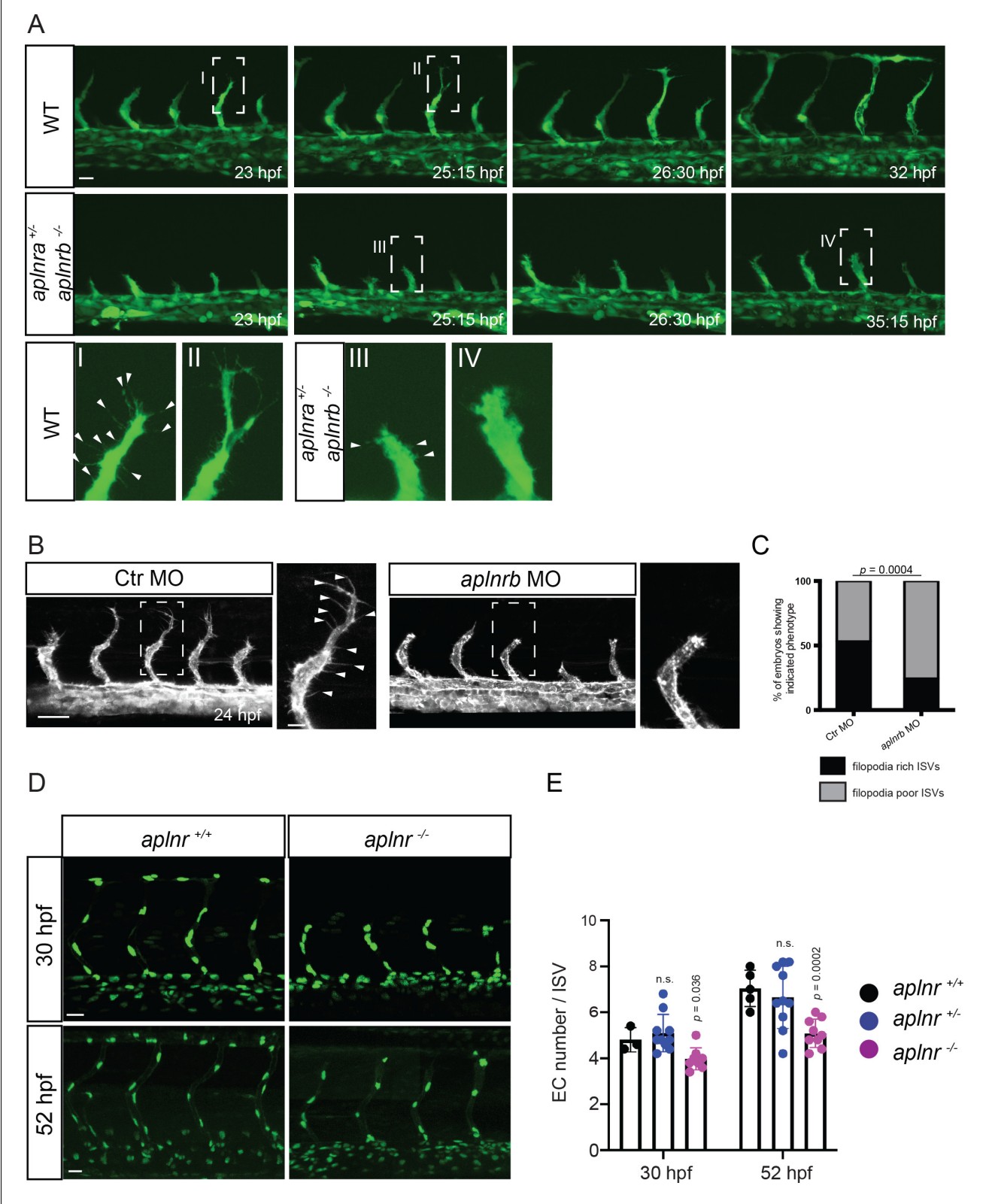

**Figure 2.** Apelin signaling regulates endothelial filopodia formation and endothelial cell numbers. (**A**) Still images from confocal time-lapse movies of vascular development in wild-type and *aplnra* [+/-]; *aplnrb* [-/-] embryos. During sprouting, wild-type tip cells send out filopodia (arrowheads). *aplnra* [+/-]; *aplnrb* [-/-] embryos exhibit smaller sprouts and fail to form filopodia. (**B**) Confocal images of the blood vasculature in 24 hpf *Tg(kdrl:HsHRAS-EGFP)* embryos injected with Ctr MO and *aplnrb* MO. *aplnrb* morphant embryos exhibit smaller sprouts and fail to form filopodia (arrowheads). (**C**) *aplnrb*

*Figure 2 continued on next page*

*Figure 2 continued*

morphant embryos exhibit a reduction in the number of endothelial filopodia (Ctr MO, n = 10; *aplnrb* MO, n = 15). (D) Confocal images of the blood vasculature of 30 and 52 hpf *Tg(fli1a:nEGFP)* wild-type and *aplnra* $^{+/-}$; *aplnrb* $^{-/-}$ embryos showing EC cell nuclei. (E) *aplnra* $^{+/-}$; *aplnrb* $^{-/-}$ embryos exhibit reduced EC numbers in the ISVs (30 hpf: *aplnr +/+*, n = 3; *aplnr +/-*, n = 10; *aplnr -/-*, n = 8; 52 hpf: *aplnr +/+*, n = 5; *aplnr +/-*, n = 10; *aplnr -/-*, n = 9). n.s. not significant (two-tailed t-test). Scale bars: A, D, 20 μm; B, 40 μm; B, inset 10 μm.

The online version of this article includes the following video and figure supplement(s) for figure 2:

**Figure supplement 1.** No obvious defects during initiation of EC sprouting in Apelin signaling-deficient embryos.

**Figure supplement 2.** No obvious defects in EC polarity in Apelin deficient embryos.

**Figure supplement 3.** Overexpression of *apln* does not cause ectopic EC sprouting.

**Figure 2—video 1.** Confocal time-lapse-imaging of a *Tg(fli1a:EGFP)* wild-type embryo from 23 to 32 hpf.

https://elifesciences.org/articles/55589#fig2video1

**Figure 2—video 2.** Confocal time-lapse-imaging of a *Tg(fli1a:EGFP) aplnra* $^{+/-}$; *aplnrb* $^{-/-}$ embryos from 23 to 35 hpf.

https://elifesciences.org/articles/55589#fig2video2

elongation was slower in these mutant embryos, resulting in an overall reduction of sprout length (*Figure 2A*, *Figure 2—video 1*, *Figure 2—video 2*). Furthermore, while endothelial tip cells in wild-type embryos formed long filopodia which extended toward the dorsal side of the animal (*Figure 2A* I, II, *Figure 2—video 1*), *aplnr* mutant embryos (*aplnra+/-; aplnrb -/-* and *aplnra -/-; aplnrb -/-*) as well as *aplnrb* morpholino (MO) injected embryos (morphants) displayed a blunted tip cell morphology (*Figure 2A* III, 2A IV, 2B, C, *Figure 2—video 2*), a phenotype which did not recover over time.

Previously, we reported a role for Apelin signaling in establishing blood flow-induced EC polarity (*Kwon et al., 2016*). To determine whether the observed defects during sprouting were caused by defects in EC polarity, we analyzed the location of the Golgi apparatus during ISV formation in wild-type and mutant embryos. However, we could not detect obvious differences in EC polarity during angiogenic sprouting (*Figure 2—figure supplement 2*, arrowheads point to polarized ECs). Next, we asked whether Apelin signaling regulates the number of ECs, and so combined *aplnr* mutants with the *Tg(fli1a:nEGFP)* reporter line (*Roman et al., 2002*) to visualize EC nuclei. Compared to controls, *aplnr* mutants exhibited a reduction in ISV EC numbers of 1 cell at 30 hpf (4 instead of 5) and 2 cells at 52 hpf (5 instead of 7) (*Figure 2D,E*). We next assessed whether *apln* overexpression leads to ectopic sprouting. To this end, we generated an inducible transgenic line to overexpress *apln* under the control of the *hsp70l* promoter. However, global overexpression of *apln* did not lead to ectopic sprouting of blood vessels but led to mispatterned lymphatic vessels (*Figure 2—figure supplement 3*, arrows). Altogether, these data indicate that the angiogenic defects in Apelin signaling-deficient embryos are caused by filopodia defects and impaired cell migration. Apelin signaling also regulates the number of ECs within the ISV sprouts.

## Apelin signaling drives the sprouting behavior of ECs

We hypothesized that *aplnrb* expression (*Figure 1B*) provides an advantage for ECs to sprout. To test this hypothesis, we generated chimeric embryos using wild-type and *aplnr* deficient embryos (*Figure 3A*). Upon transplantation of wild-type donor cells into wild-type hosts, 34,5% of the donor-derived ECs were present in the ISVs at 24 hpf (*Figure 3B,C*). In contrast, upon transplantation of wild-type donor cells into *aplnr*-deficient hosts, 80% of the donor-derived ECs were present in the ISVs at 24 hpf (*Figure 3B,C*). Together these data show that the apelin receptors function cell-autonomously in endothelial sprouting. The Apelin receptor has been shown to signal mainly through the G-protein Gαi (*Habata et al., 1999*). Therefore, we blocked Gαi function through the mosaic and vascular-specific overexpression of pertussis toxin (PTX). Our results show that ECs deficient for signaling though Gαi behave similarly to *aplnr* mutant ECs indicating that the Apelin receptor mediates its angiogenic effect through Gαi (*Figure 3—figure supplement 1*). Notably, wild-type donor-derived ECs in *aplnr* deficient embryos populated the entire dorsal part of the vasculature which is usually missing in these mutants, further confirming the cell-autonomous function of the Apelin receptors during angiogenesis (*Figure 3—figure supplement 2*). Together, these results indicate that apelin signaling primes ECs toward a sprouting state.

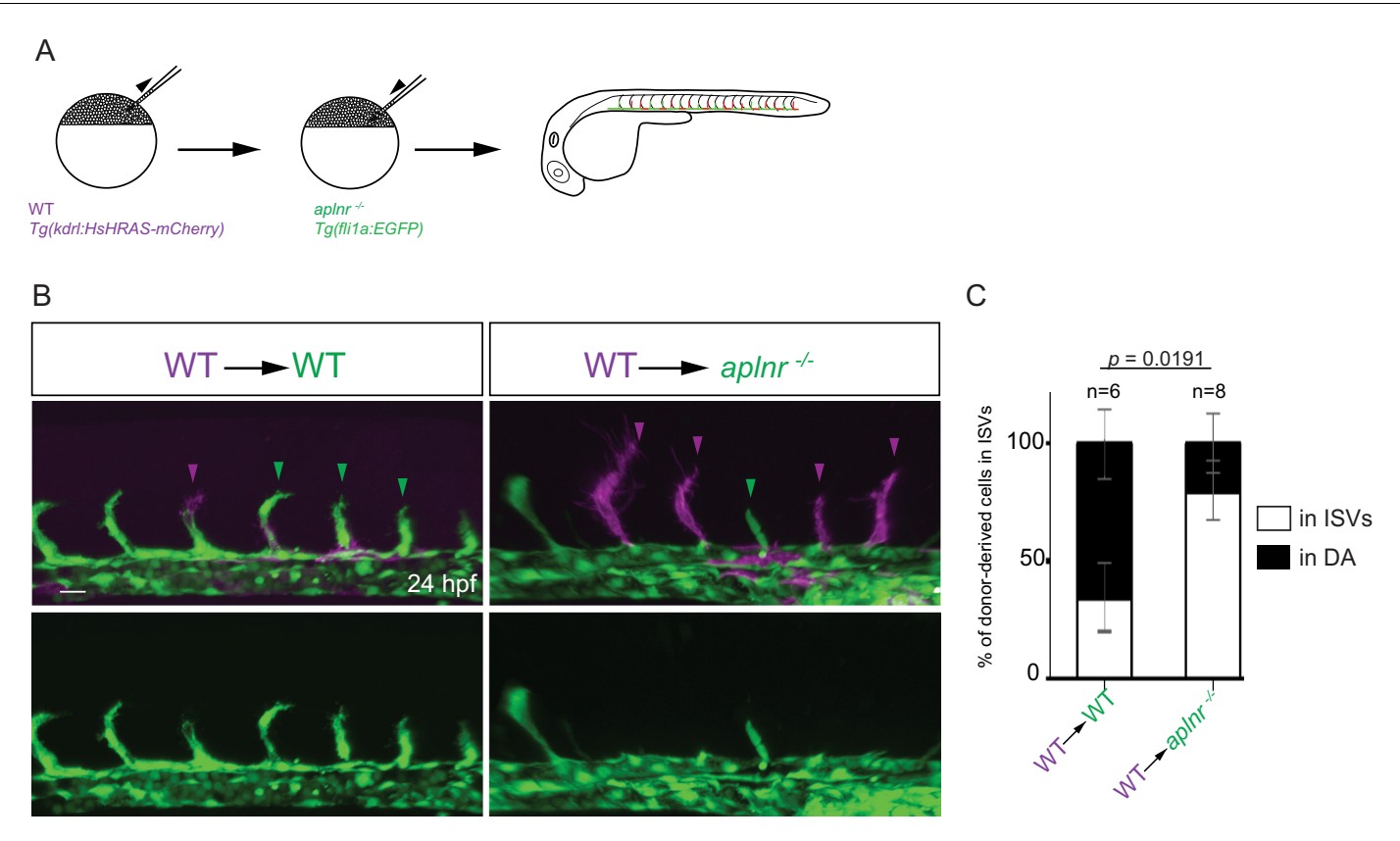

**Figure 3.** Apelin signaling promotes the sprouting behavior of ECs. (**A**) Experimental design: At the blastula stage, cells from *Tg(kdrl:HsHRAS-mCherry)* embryos were transplanted into host embryos obtained from *Tg(fli1a:EGFP) aplnra* $^{+/-}$; *aplnrb* $^{+/-}$ incrosses. At 24 hpf, the mosaic embryos were imaged and the donor-derived ECs scored for their position. (**B, C**) 34,5% of wild-type donor-derived ECs in wild-type hosts were found within the ISVs. 80% of wild-type donor-derived ECs in *aplnra* $^{+/-}$; *aplnrb* $^{-/-}$ hosts were found within the ISVs. Notably, wild-type ECs transplanted into *aplnr-* deficient embryos completely substituted for the lack of cells in the dorsal part of the vasculature at 54 hpf (*Figure 3—figure supplement 2*). Scale bars: B, 20 μm.

The online version of this article includes the following figure supplement(s) for figure 3:

**Figure supplement 1.** Overexpression of PTX phenocopies loss of Apelin signaling.
**Figure supplement 2.** Apelin signaling functions cell-autonomously in ECs.

## Apelin signaling functions downstream of Notch signaling

It has been previously reported that Notch-deficient ECs outcompete wild-type ECs during ISV sprouting (*Siekmann and Lawson, 2007*), an observation consistent with data in mouse (*Jakobsson et al., 2010*; *Pitulescu et al., 2017*). Since wild-type ECs similarly outcompete *aplnr* mutant ECs, we wanted to investigate potential links between Apelin and Notch signaling. Hence, we first blocked Notch signaling in *Tg$^{BAC}$(apln:EGFP)* embryos by injecting a *dll4* MO. As previously reported (*Leslie et al., 2007*; *Siekmann and Lawson, 2007*), *dll4* morphants exhibited a hypersprouting ISV phenotype (*Figure 4A*). Notably, we also observed a clear increase in *apln:EGFP* expression in the ectopic sprouts (*Figure 4A*). To test whether Apelin signaling is required as a downstream effector of Notch signaling during angiogenesis, we injected the *dll4* MO into the offspring of *apln* and *aplnrb* heterozygous parents and compared the phenotype in homozygous mutant embryos versus their wild-type siblings. Strikingly, the hypersprouting phenotype of *dll4* morphants was not present when Apln or Aplnrb function was lost (*Figure 4B*, *Figure 4—figure supplement 1*). To examine whether other hypersprouting phenotypes require Apelin signaling, we analyzed *aplnrb, plexinD1 (plxnd1)* double mutant embryos (*Figure 4C,E*). *aplnrb* mutant embryos exhibit reduced sprouting and *plxnd1* mutant embryos exhibit ectopic sprouting in line with published data (*Torres-Vázquez et al., 2004*; *Figure 4C,E*). Loss of *aplnrb* function in the background of the *plxnd1* mutant did not alter its hypersprouting phenotype (*Figure 4C,E*), suggesting that it is

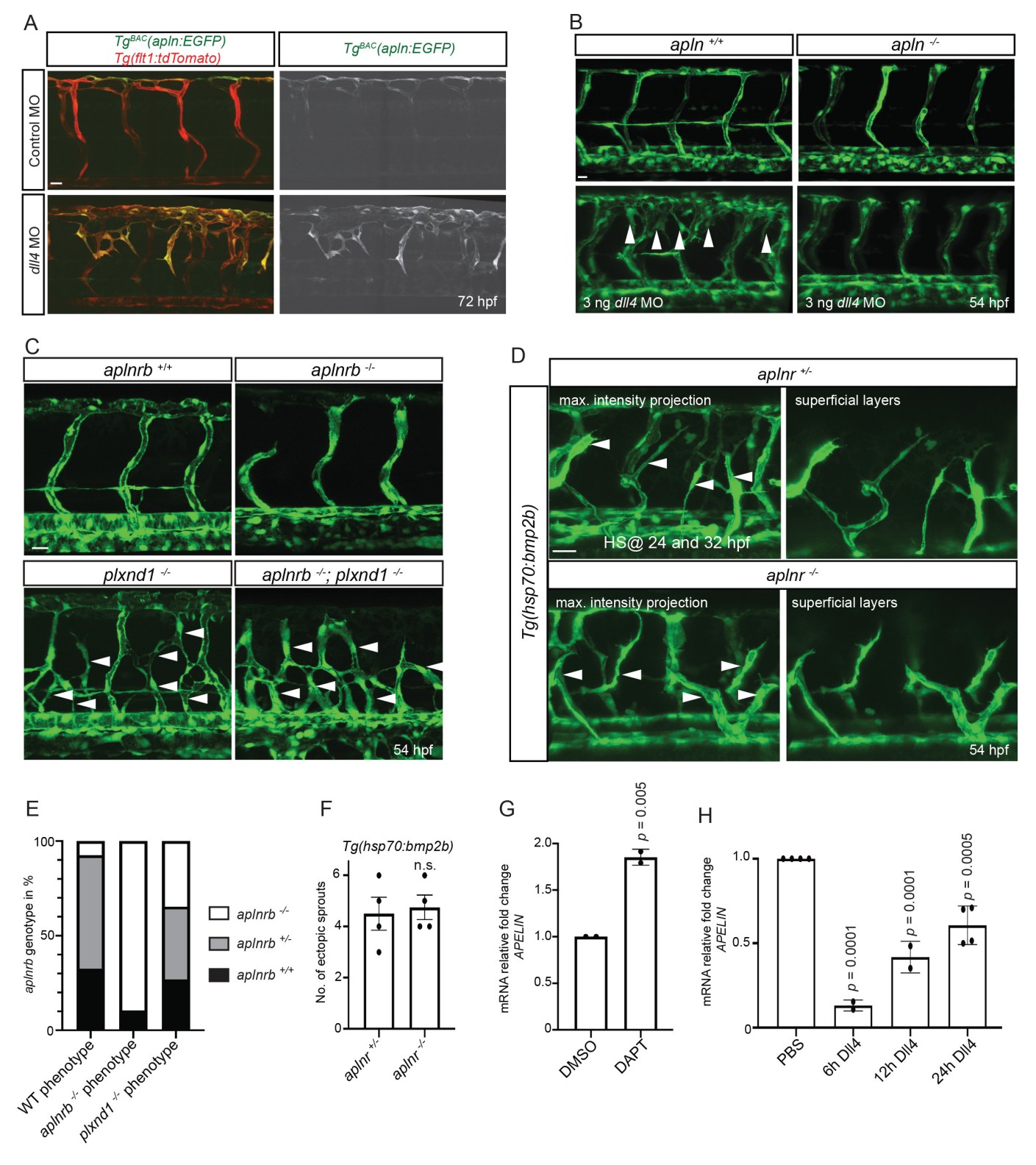

**Figure 4.** Apelin signaling functions downstream of Notch signaling in endothelial cells. (A - D) Confocal projection images of the blood vasculature in the trunk region of *Tg(flt1:tdTomato)* (A) and *Tg(fli1a:EGFP)* (B–D) animals at 54 (B–D) and 72 (A) hpf. (A) Injection of a *dll4* morpholino leads to an increase in *Tg^BAC^(apln*:EGFP) expression. (B) Loss of Apelin function can block excessive endothelial sprouting in *dll4* morphants. (C, E) Angiogenic response in *aplnrb^-/-^*, *plxnd1^-/-^*, and *aplnrb^-/-^; plxnd1^-/-^* embryos (arrowheads) (n = 95). (D, F) Angiogenic response to *bmp2b* overexpression in *aplnr*

*Figure 4 continued on next page*

*Figure 4 continued*

[+/-] and *aplnr* [-/-] embryos (arrowheads). (**E**) Genotype of embryos for *aplnrb* after sorting them according to phenotype. (**G**) RT-qPCR analysis of *APELIN* mRNA levels in HUVECs treated with DAPT for 24 hr. Blocking Notch signaling with DAPT induces *APELIN* expression. (**H**) RT-qPCR analysis of *APELIN* mRNA levels in HUVECs cultured on DLL4 to activate Notch signaling. Activating Notch signaling represses *APELIN* expression. Arrowheads point to ectopic sprouts. n.s. not significant (two-tailed t-test). Ct values can be found in *Figure 4—source data 1*. Scale bars: A, C, 20 μm; B, 15 μm; D, 30 μm. The online version of this article includes the following source data and figure supplement(s) for figure 4:

**Source data 1.** Ct values of RT-qPCR.
**Figure supplement 1.** Apelin signaling functions downstream of Notch signaling in ECs.

independent of Apelin function. Similar results were obtained when we analyzed ectopic venous sprouting in response to *bmp2b* overexpression (*Wiley et al., 2011*; *Figure 4D,F*). Together these data indicate that Apelin signaling is specifically required for Notch-modulated angiogenesis. To investigate whether *apln* expression is itself regulated by Notch signaling, we performed cell culture assays. We treated HUVECs with the Notch inhibitor DAPT and analyzed *APLN* expression by RT-qPCR. Consistent with the observations in zebrafish (*Figure 4A*), we observed an increase in *APLN* expression upon Notch inhibition (*Figure 4G*, *Figure 4—source data 1*). Next, we activated Notch signaling by stimulating HUVECs with the Notch ligand DLL4. Conversely to the Notch inhibition data, activating Notch signaling in HUVECs suppressed *APLN* expression (*Figure 4H*, *Figure 4— source data 1*). Together these data suggest that the increased sprouting in response to Notch inhibition is, in part, driven by the upregulation of *apln*.

## Apelin signaling positively regulates EC metabolism

Because EC sprouting requires an increase in metabolic activity (*Dobrina and Rossi, 1983*; *Krützfeldt et al., 1990*; *Mertens et al., 1990*; *De Bock et al., 2013a*; *Vandekeere et al., 2015*) and Apelin signaling has been shown to control cell metabolism in other contexts (*Dray et al., 2008*; *Sawane et al., 2013*), we asked whether Apelin signaling promotes EC metabolism. Previous studies have demonstrated that ECs rely on glycolysis for sprouting (*De Bock et al., 2013a*; *Vandekeere et al., 2015*). Therefore, we measured the extracellular acidification rate (ECAR) as a surrogate parameter of glycolysis in Apelin signaling-deficient HUVECs (*Figure 5A,B*). Notably, we observed a marked reduction in glycolysis after knockdown of Apelin signaling (*Figure 5A*), whereas mitochondrial oxygen consumption appeared unchanged (*Figure 5B*). To gain insight into the underlying mechanisms, we analyzed key regulators of metabolism and found a reduction in c-MYC protein levels after depletion of Apelin signaling (*Figure 5C*). Furthermore, expression of *PFKFB3*, which encodes an enzyme that sustains high glycolytic rates, was also reduced in Apelin signaling-deficient HUVECs (*Figure 5D*). In order to analyze whether a reduction in EC metabolic activity causes the vascular phenotype observed in *aplnrb* mutants, we performed mosaic rescue experiments and overexpressed *pfkfb3* in ECs. In agreement with our in vitro data, we found that overexpression of *pfkfb3* in endothelial tip cells leads to a partial rescue of the vascular phenotype in *aplnrb* mutants (*Figure 5E* arrowheads, 5F, *Figure 4—source data 1*). Thus, Apelin signaling controls the expression of regulators of glucose metabolism as well as glycolytic activity in developing endothelial cells.

## Discussion

During the formation of the first embryonic blood vessels, angioblasts migrate to the midline where they coalesce to form the future DA and cardinal vein. We have previously reported that vasculogenesis relies on the function of the ligand Apela (*Helker et al., 2015*). Here, we show that angiogenesis depends mostly on the function of the ligand Apln. However, Apela can partially compensate for the loss of Apln. This stage-specific ligand usage is in agreement with previous studies showing that *apela* expression is reduced by the end of vasculogenesis when *apln* starts to be expressed (*Chng et al., 2013*; *Pauli et al., 2014*).

During angiogenesis in embryos lacking Apelin signaling, we observed a severe sprouting defect with a reduction in EC numbers and filopodia. As ECs proliferate, extend filopodia, and migrate during ISV formation, it is challenging to assign the cause of the sprouting defect to the EC proliferation or filopodia formation defects. However, *Phng et al., 2013* reported that the inhibition of filopodia

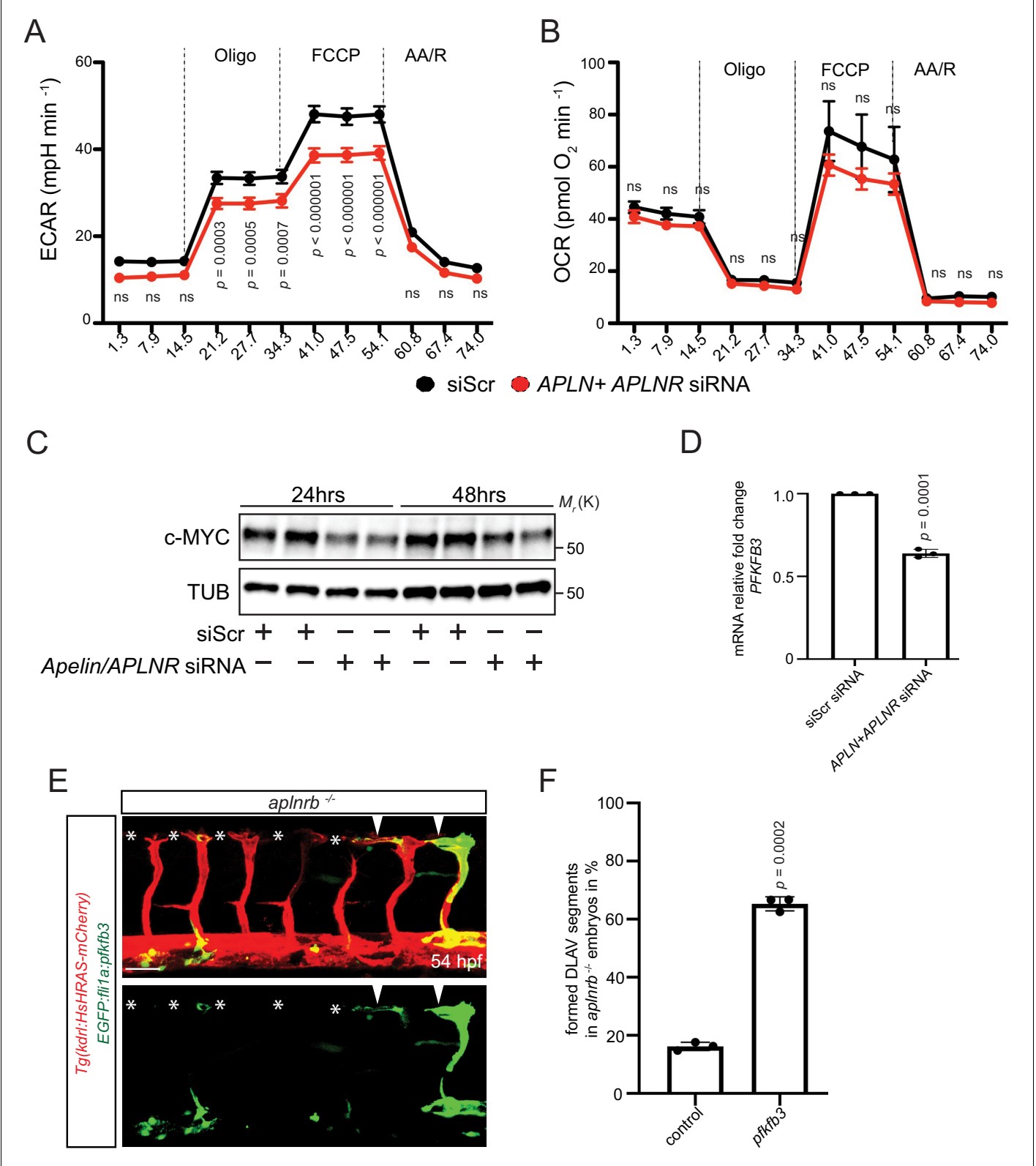

**Figure 5.** Apelin signaling positively regulates EC metabolism. (A - B) Extracellular acidification aate (ECAR) (**A**) and oxygen consumption rates (OCR) (**B**) in siScr and *APLN+APLNR* siRNA-treated HUVECs under basal conditions and in response to oligomycin, fluoro-carbonyl cyanide phenylhydrazone (FCCP) and antimycin A (AA)/rotenone. (**A**) Reduced basal and maximal glycolytic activity in *APLN+APLNR* siRNA-treated compared to siScr-treated HUVECs. (**B**) No significant difference in oxygen consumption in *APLN+APLNR* siRNA-treated compared to siScr-treated HUVECs. (**C**) Reduced c-MYC

*Figure 5 continued on next page*

*Figure 5 continued*

levels in *APLN+APLNR* siRNA-treated compared to siScr-treated HUVECs. (D) RT-qPCR analysis of *PFKFB3* mRNA levels in *APLN+APLNR* siRNA-treated compared to siScr-treated HUVECs. (E) Confocal projection images of the blood vasculature in the trunk region of a 54 hpf *Tg(kdrl:HsHRAS-mCherry)* animal injected with an *EGFP:fli1a:pfkfb3* plasmid. Arrowheads point to formed DLAV fragments while asterisks indicate missing DLAV fragments. (F) Quantification of the rescue of the DLAV fragment by mosaic *pfkfb3* overexpression in *aplnrb* $^{-/-}$ embryos. n.s. not significant (two-tailed t-test). Scale bar: E, 50 μm.

The online version of this article includes the following figure supplement(s) for figure 5:

**Figure supplement 1.** Schematic model.

formation by Latrunculin B treatment reduces ISV sprout length, suggesting that the ISV sprouting defects in *apln* mutants is caused by the filopodia defects. However, one cannot exclude the possibility that defects in EC numbers are also contributing to the ISV sprouting defects.

While we observed a severe angiogenesis phenotype when Apelin signaling was impaired, global overexpression of *apln* did not lead to ectopic sprouting. However, these experiments were done in the presence of endogenous Apelin, and thus, it is possible that the endogenous Apelin gradient prevents ECs from ectopic sprouting. In addition, Apelin might need to be expressed from a discrete source, rather than globally, to elicit a sprouting response.

During sprouting angiogenesis, ECs within a sprout are highly heterogenous in their shape, gene expression and function, which led to the model of tip and stalk cells (*Gerhardt et al., 2003*). While differences in expression between tip and stalk cells have been reported for several genes (*Tammela et al., 2008*), (*Hellström et al., 2007*; *Siekmann and Lawson, 2007*; *Leslie et al., 2007*; *Suchting et al., 2007*; *del Toro et al., 2010*; *Bussmann et al., 2011*; *Herbert et al., 2012*), little is known about the molecular differences between sprouting and resting ECs (*Schlereth et al., 2018*). By analyzing novel reporter lines for *apln* and *aplnrb* expression, we observed high *apln* expression in tip cells while we could not observe any difference in *aplnrb* expression between tip and stalk cells (*Figure 5—figure supplement 1*). Interestingly, *aplnrb* is highly expressed in sprouting ECs in ISVs while being absent from non-angiogenic ECs in the DA (*Figure 5—figure supplement 1*). These observations are in line with a recent study showing that ECs during tumor angiogenesis can be labeled by a *CreERT2* transgene in the *Aplnr* locus while quiescent blood vessels in the surrounding tissue are not labeled (*Zhao et al., 2018*).

At the molecular level, vascular sprouting and cell positioning within the sprout is tightly regulated by VEGF and Notch signaling (*Hellström et al., 2007*; *Lobov et al., 2007*; *Siekmann and Lawson, 2007*; *Suchting et al., 2007*; *Jakobsson et al., 2010*). In addition to these signaling pathways, we propose Apelin signaling as a molecular switch to drive ECs into a pro-angiogenic state. In line with the expression of *aplnrb* in sprouting but not quiescent ECs, we show that *aplnrb* function regulates the ability of ECs to sprout or stay quiescent. Similarly, Notch signaling regulates the behavior of ECs (*Siekmann and Lawson, 2007*): *rbpj* deficient ECs contribute to the ISVs while wild-type ECs stay within the DA (*Siekmann and Lawson, 2007*). Of note, we found that Notch signaling regulates the expression of *apln* in vitro as well as in vivo and that Apelin signaling is a key downstream effector of Notch signaling during angiogenesis (*Figure 5—figure supplement 1*). However, it is very unlikely that *apln* is a direct Notch target gene since activation of Notch signaling leads to a downregulation of *APLN* expression. Thus far, two downstream effectors of Notch signaling have been reported to control angiogenesis namely PTEN (*Serra et al., 2015*) and CXCR4, another GPCR (*Hasan et al., 2017*; *Pitulescu et al., 2017*). While PTEN has been shown to be required for Notch induced arrest in EC proliferation (*Serra et al., 2015*), CXCR4 mediates Notch-controlled EC migration (*Hasan et al., 2017*; *Pitulescu et al., 2017*). PTEN and Apelin both regulate AKT phosphorylation (*Davies et al., 1998*; *Masri et al., 2004*). Thus, one might speculate that AKT function is a common effector of PTEN and Apelin signaling in EC proliferation. Furthermore, we found that Apelin was required for EC migration in the absence of Notch signaling. Similarly, CXCR4 is required for EC migration in the absence of Notch signaling (*Hasan et al., 2017*; *Pitulescu et al., 2017*). CXCR4 and APLNR both signal through the G-protein Gαi (*Moepps et al., 1997*; *Habata et al., 1999*), and they might therefore have similar effects. Gpr124, another GPCR, has been reported to be required in tip cells during zebrafish angiogenesis (*Vanhollebeke et al., 2015*), similar to Aplnr. However,

Gpr124 is required in tip cells only in the brain (*Vanhollebeke et al., 2015*), while Aplnr is required in tip cells in the ISVs, where it is most highly expressed.

Sprouting angiogenesis is controlled by genetically encoded signal transducers as well as by the metabolic state. However, how environmental signals modulate the metabolic activity of ECs is incompletely understood. Here, we show that Apelin signaling regulates the expression of *PFKFB3* and c-MYC, two powerful drivers of EC metabolism (*Wilhelm et al., 2016*; *De Bock et al., 2013b*). Recently it has been shown that Apelin signaling promotes FOXO1 phosphorylation (*Hwangbo et al., 2017*), which negatively regulates its activity. Consistent with these findings, FOXO1 has been shown to suppress c-MYC expression (*Wilhelm et al., 2016*). Together these data raise the possibility that Apelin signals through FOXO1 to regulate c-MYC levels. Of note, genetic deletion of *Pfkfb3* in mouse ECs leads to a reduction in their number as well as defects in filopodia formation and extension (*De Bock et al., 2013b*), phenocopying *aplnr* mutant embryos.

Taken together, our findings provide novel insights into a druggable pathway regulating angiogenesis and suggest that manipulating the angiogenic state of ECs by controlling Apln signaling might have therapeutic potential to control vascular growth in pathological settings.

# Materials and methods

## Key resources table

| Reagent type (species) or resource | Designation | Source or reference | Identifiers | Additional information |
|---|---|---|---|---|
| genetic reagent (*D. rerio*) | Tg(fli1a:EGFP)$^{y1}$ | *Lawson and Weinstein, 2002* | ZFIN: *y1* | |
| genetic reagent (*D. rerio*) | Tg(fli1a:nEGFP)$^{y7}$ | *Roman et al., 2002*, | ZFIN: *y7* | |
| genetic reagent (*D. rerio*) | Tg(kdrl:HsHRAS-mCherry)$^{s896}$ | *Chi et al., 2008* | ZFIN: *s896* | |
| genetic reagent (*D. rerio*) | aplnra$^{mu296}$ | *Helker et al., 2015* | ZFIN: *mu296* | |
| genetic reagent (*D. rerio*) | aplnrb$^{mu281}$ | codes for another allele of *aplnrb* from *Helker et al., 2015* | ZFIN: *mu281* | |
| genetic reagent (*D. rerio*) | apln$^{mu267}$ | *Helker et al., 2015* | ZFIN: *mu267* | |
| genetic reagent (*D. rerio*) | Tg(hsp 70:bmp2b)$^{fr13}$ | *Chocron et al., 2007* | ZFIN: *fr13* | |
| genetic reagent (*D. rerio*) | apela$^{br13}$ | *Chng et al., 2013*, | ZFIN: *br13* | |
| genetic reagent (*D. rerio*) | Tg(fli:lifeact-GFP)$^{mu240}$ | *Hamm et al., 2016* | ZFIN: *mu240* | |
| genetic reagent (*D. rerio*) | Tg(fli1a:Hsa.B4GALT1-mCherry)$^{bns9}$ | *Kwon et al., 2016* | ZFIN: *bns9* | |
| genetic reagent (*D. rerio*) | Tg(hsp70:apln)$^{mu269}$ | This manuscript | ZFIN: *mu269* | |

*Continued on next page*

*Continued*

| Reagent type (species) or resource | Designation | Source or reference | Identifiers | Additional information |
|---|---|---|---|---|
| genetic reagent (*D. rerio*) | *Tg(kdrl:HsHRAS-EGFP)^mu280* | This manuscript | ZFIN: *mu280* | |
| genetic reagent (*D. rerio*) | *Tg(apln:EGFP)^bns157* | This manuscript | ZFIN: *bns157* | |
| genetic reagent (*D. rerio*) | *Tg(aplnrb: aplnrb-TagRFP-sfGFP)^bns309* | This manuscript | ZFIN: *bns309* | |
| antibody | anti-FOXO1 (rabbit monoclonal) | Cell Signaling Technology | Cat#2880 | (1:1000) |
| antibody | anti-pThr24FOXO1/pThr32FOXO3a (rabbit monoclonal) | Cell Signaling Technology | Cat#9464 | (1:1000) |
| antibody | anti-c-MYC (rabbit polyclonal) | Cell Signaling Technology | Cat#9402 | (1:1000) |
| antibody | anti-Tubulin (rabbit polyclonal) | Cell Signaling Technology | Cat#2148 | (1:1000) |
| other | Taqman probe *APLN* | Thermo Fisher Scientific | Hs00175572_m1 | |
| other | Taqman probe *APLNR* | Thermo Fisher Scientific | Hs00270873_s1 | |
| other | Taqman probe *PFKFB3* | Thermo Fisher Scientific | Hs00270873_s1 | |
| other | Taqman probe *ACTB* | Thermo Fisher Scientific | Hs01060665_g1 | |
| commercial assay or kit | In-Fusion HD Cloning Plus | Takara Bio | Cat# 638910 | |
| transfected construct (human) | *APLN* | Dharmacon | Cat# L-017023-01-0005 | 50 nM |
| transfected construct (human) | *APLNR* | Dharmacon | Cat# L-005430-00-0005 | 50 nM |
| transfected construct (human) | ON-TARGETplus Non-targeting Pool | Dharmacon | Cat# D-001810-10-05 | 50 nM |
| commercial assay or kit | mMessage mMachine SP6 Transcription Kit | Thermo Fisher Scientific | Cat# AM1340 | |
| commercial assay or kit | DIG RNA labelling kit | Roche | Cat# 11277073910 | |
| commercial assay or kit | SuperScript III First-Strand Synthesis System | Thermo Fisher Scientific | Cat#18080051 | |
| commercial assay or kit | RNA Clean and Concentrator Kit | Zymo Research | Cat# R1013 | |
| software, algorithm | ZEN Blue 2012 | Zeiss, Germany | | |

*Continued on next page*

*Continued*

| Reagent type (species) or resource | Designation | Source or reference | Identifiers | Additional information |
|---|---|---|---|---|
| software, algorithm | ZEN Black 2012 | Zeiss, Germany | | |
| software, algorithm | Imaris - Version 8.4.0 | Bitplane, UK | | |
| software, algorithm | GraphPad Prism 6 | GraphPad Software, USA | | |

## Zebrafish husbandry and strains

All zebrafish housing and husbandry were performed under standard conditions in accordance with institutional (Max Planck Society) and national ethical and animal welfare guidelines approved by the ethics committee for animal experiments at the Regierungspräsidium Darmstadt, Germany, as well as the FELASA guidelines (*Aleström et al., 2020*). Embryos were staged by hours post fertilization (hpf) at 28.5°C (*Kimmel et al., 1995*). The following lines were used: *Tg(fli1a:EGFP)^y1^* (*Lawson and Weinstein, 2002*), *Tg(fli1a:nEGFP)^y7^* (*Roman et al., 2002*), *Tg(kdrl:HsHRAS-mCherry)^s896^* (*Chi et al., 2008*), *aplnra^mu296^* (*Helker et al., 2015*), the *aplnrb^mu281^* allele was generated using the same CRISPR as in *Helker et al., 2015* and contains a 4 bp insertion 137 bp downstream of the ATG leading to a premature stop codon 196 bp downstream of the ATG, *apln^mu267^* (*Helker et al., 2015*), *Tg(hsp70:bmp2b)^fr13^* (*Chocron et al., 2007*), *apela^br13^* (*Chng et al., 2013*), *Tg(fli1a:LIFEACT-GFP)^mu240^* (*Hamm et al., 2016*), *Tg(fli1a:Hsa.B4GALT1-mCherry)^bns9^* (*Kwon et al., 2016*), *Tg(hsp70:apln)^mu269^* (this study), *Tg(kdrl:HsHRAS-EGFP)^mu280^* (this study), *Tg(apln:EGFP)^bns157^* (this study) and *Tg(aplnrb:aplnrb-TagRFP-sfGFP)^bns309^* (this study).

## Generation of the *Tg^BAC^(apln:EGFP)^bns157^*, *Tg^BAC^(aplnrb:aplnrb-TagRFP-sfGFP)^bns309^*, *Tg(kdrl:HsHRAS-EGFP)^mu280^,* and *Tg(hsp70l:apln)^mu269^* lines

To generate the *apln* and *aplnrb* bacterial artificial chromosome (BAC) constructs, we used the BAC clones RP71-2G21 containing the *apln* locus and CH211-102K containing the *aplnrb* locus. All recombineering steps were performed as described in *Bussmann and Schulte-Merker, 2011* with the modifications as described in *Helker et al., 2019*. The following homology arms were used to generate the targeting PCR products of the EGFP_Kan, and TagRFP-sfGFP_Kan cassettes:*apln*-HA1: 5'-ccactacagtatatcagctagcgactggcagggaaacggaggggagagcaaccatggtgagcaagggcgaggag-3' and *apln*-HA2: 5'-cacagcagagaaaccaccagcacaatcaccagcgtcaagatcttcacattttccagaagtagtgaggag-3';*aplnrb*-HA1:5'-gctccctttcttcacagaagaccgaggcccagtcgctggctacgaaggtgcttggacctggactcggatc-'3 and *aplnrb*-HA2: 5'-taattgctgacttgttaccccaattctgcgtcacccttccgttctcctcctgaccatgattacgccaagc-'3.

To generate the *Tg(kdrl:HsHRAS-EGFP)* and *Tg(hsp70:apln) lines,* the gateway recombination system (Invitrogen) using entry vectors and the pTolDest destination vector (*Villefranc et al., 2007*) was used. The *apln* coding sequence was amplified from cDNA. 100 pg DNA of the plasmids and 50 pg of tol2 mRNA were injected into one-cell stage zebrafish embryos for stable germline transmission.

## Morpholino injections

Morpholinos were obtained from Gene Tools, resuspended in distilled $H_2O$ and around 2 nl was injected into 1 cell stage embryos. The following morpholinos were used: *aplnrb* MO (*Helker et al., 2015*) at 0.5 ng/embryo, *dll4* MO (*Hogan et al., 2009*) at 3 ng/embryo. An equal amount of the standard control MO: 5'-CCTCTTACCTCAGTTACAATTTATA-3' was used for each experiment.

## Transplantation experiments

At the sphere stage, cells were removed from *Tg(kdrl:HsHRAS-mCherry)* donor embryos and transferred to *Tg(fli1a:EGFP) aplnr* mutant hosts using a glass capillary. Transplanted ECs were identified by transgenic mCherry expression.

## Whole-mount in situ hybridization

Single in situ hybridizations were performed as described (*Thisse and Thisse, 2008*; *Helker et al., 2013*). The following probes were synthesized: *apln* (*Helker et al., 2015*), *apela* (*Chng et al., 2013*), *aplnra* (*Helker et al., 2015*), and *aplnrb* (*Helker et al., 2015*).

## Confocal microscopy

Zebrafish larvae were mounted in 1% low melt agarose. Egg water and agarose were supplemented with 19.2 mg/l Tricaine. All fluorescent images were acquired using an upright Zeiss LSM 780, 800 or 880 or a Leica SP5 or SP8 confocal microscope. Maximum projection images were analyzed and generated using Imaris (Bitplane).

## Quantification of mutant phenotypes

For every embryo, somites 5 to 15 were analyzed (normal: 10 fully developed ISVs and connected DLAV; mild: 10 ISVs fully developed but no DLAV; strong: 1 to 6 ISVs shortened; severe: 1 to 10 ISVs shortened).

## Quantification of filopodia

Only filopodia with more than 10 µm in length were used for quantification.

ISVs were categorized as filopodia rich ISVs (more than six filopodia) or filopodia poor ISVs (less than six filopodia). A total of 49 ISVs were quantified for the Ctr MO and 103 ISVs for the *aplnrb* MO.

## *pfkfb3* rescue experiments

*pfkfb3* was cloned downstream of a bidirectional *fli1a* promoter driving *EGFP* in one direction and *pfkfb3* in the other direction. *aplnrb* mutant embryos were injected with 20 pg *EGFP:fli1a:pfkfb3* DNA and 30 pg Tol2 mRNA to generate mosaic blood vessels. EGFP positive tip cells were analyzed to quantify the percentage of connected DLAV segments. Neighboring EGFP negative tip cells were used as controls.

## Cell culture

Pooled human umbilical vein endothelial cells (HUVECs) were purchased from Lonza (#CC-2519) and cultured in endothelial basal medium (EBM, Lonza) supplemented with hydrocortisone (1 µg/ml), bovine brain extract (12 µg/ml), gentamicin (50 µg/ml), amphotericin B (50 ng/ml), epidermal growth factor (10 ng/ml), and 10% fetal bovine serum (FBS, Life Technologies). HUVECs were tested for mycoplasma and cultured until the fourth passage. Cells were maintained at 37°C in a humidified atmosphere with 5% $CO_2$.

## RNA interference

To silence *APLN* and *APLNR* gene expression, HUVECs were transfected with 50 nM APLN and APLNR ON-TARGET SMARTpool siRNA (Dharmacon). As a control, a non-targeting siRNA pool was used (Dharmacon). HUVECs were grown to 70% confluency and transfected with Lipofectamine RNAiMAX (Life Technologies) according to manufacturer's instructions.

## Western blot analysis and antibodies

Western blot analyses were performed with precast gradient gels (Bio-Rad) using standard methods. Briefly, cells were lysed in RIPA buffer (Sigma; 150 mM NaCl, 1.0% IGEPAL CA-630, 0.5% sodium deoxycholate, 0.1% SDS, and 50 mM Tris, pH 8.0) supplemented with Complete Protease Inhibitor Cocktail (Roche) and 1 mM PMSF. Proteins were separated by SDS-PAGE (Tris-glycine gels with Tris/glycine/SDS buffer, Bio-Rad) and transferred onto nitrocellulose membranes using the Trans Turbo Blot system (Bio-Rad). Membranes were probed with specific primary antibodies and then with peroxidase-conjugated secondary antibodies. The following primary antibodies were used: FOXO1 (Cell Signaling Technology, #2880, 1:1000), pThr24FOXO1/pThr32FOXO3a (Cell Signaling Technology, #9464, 1:1000), c-MYC (Cell Signaling Technology, #9402, 1:1000), Tubulin (Cell Signaling Technology, #2148, 1:1000), Secondary antibodies are peroxidase-conjugated Goat IgGs (1:5000) purchased from Jackson Immuno Research Labs. The target proteins were visualized by chemiluminescence

using an ECL detection kit (Clarity Western ECL Substrate, Bio-Rad) and a ChemiDoc MP Imaging System (Bio-Rad).

## RT-qPCR

Total RNA from HUVECs was extracted using a RNeasy Mini Kit (Qiagen). Reverse transcription polymerase chain reaction (RT-PCR) was performed using a SuperScript III First-Strand Synthesis System (Invitrogen) according to manufacturer's instructions. RT-qPCR was carried out to quantify gene expression levels on a CFX connect Realtime System (Bio-Rad) with the following Taqman probes: *APLN* Hs00175572_m1, *APLNR* Hs00270873_s1, *PFKFB3* Hs00270873_s1. Each sample was normalized to the housekeeping probe *ACTB* Hs01060665_g1.

## Metabolic assay

The metabolism of cells was assessed by the measurement of extracellular acidification (ECAR) and oxygen consumption rates (OCR) using a Seahorse XFe96 analyser (Agilent). Four hours before the measurement, 40.000 HUVECs per well were seeded in a fibronectin-coated XFe96 microplate. The measurement was done following manufacturer's protocol. To monitor glycolysis, the glycolysis stress test kit was used. The following substances were sequentially injected after a baseline measurement: Glucose (10 mM), Oligomycin (3 μM) and 2-Deoxyglucose (2-DG; 100 mM). The oxygen consumption rate was assessed using the Mito stress test kit. After a baseline measurement, the following substances were sequentially injected: Oligomycin (3 μM), the mitochondrial uncoupler carbonyl cyanide-4-(trifluoromethoxy)phenyl-hydrazone (FCCP; 1 μM) as well as a mixture of antimycin A (1.5 μM) and rotenone (3 μM).

## Statistics

Standard error of the mean and P-values from a two-tailed t-test were calculated using Prism.

## Acknowledgements

We would like to thank Radhan Ramadass and the Marburg Center of Advanced Light Microscopy, Marianne Ploch and all the fish facility staff for technical support; Teja Mullapudi, Giulia Boezio, Jialing Qi and Constanze Heinzen for suggestions and critical comments on the project. Research in the Herzog laboratory was supported by the North Rhine-Westphalia 'return fellowship' and the DFG (HE4585/3-1). Research in the Potente laboratory is supported by the Max Planck Society, the European Research Council (ERC) Consolidator Grant EMERGE (773047), the DFG (SFB 834), the Excellence Cluster Cardio-Pulmonary Institute (EXC 2026, Project ID: 390649896), the DZHK (German Center for Cardiovascular Research), and a grant by the Leducq Foundation. Research in the Stainier laboratory is supported in part by the Max Planck Society, the DFG (SFB 834/4), the EU (ERC AdG project: ZMOD 694455) and the Leducq Foundation, and in the Helker laboratory by the DFG (SFB 834/4) and GRK2213.

## Additional information

### Competing interests

Didier YR Stainier: Senior editor, *eLife*. The other authors declare that no competing interests exist.

### Funding

| Funder | Grant reference number | Author |
| --- | --- | --- |
| Deutsche Forschungsgemeinschaft | SFB 834 | Christian SM Helker<br>Didier YR Stainier |
| Deutsche Forschungsgemeinschaft | GRK2213 | Christian SM Helker<br>Jean Eberlein<br>Julian Malchow |
| Deutsche Forschungsgemeinschaft | HE4585/3-1 | Wiebke Herzog |

| H2020 European Research Council | EMERGE (773047) | Michael Potente |
|---|---|---|
| Deutsche Forschungsgemeinschaft | EXC 2026 | Michael Potente |
| H2020 European Research Council | AdG project: ZMOD 694455 | Didier YR Stainier |
| North Rhine-Westphalia | Return fellowship | Wiebke Herzog |
| Max-Planck-Gesellschaft | | Christian SM Helker Kerstin Wilhelm Toshiya Sugino Hyouk-Bum Kwon Hans-Martin Maischein |

The funders had no role in study design, data collection and interpretation, or the decision to submit the work for publication.

## Author contributions
Christian SM Helker, Conceptualization, Supervision, Funding acquisition, Investigation, Visualization, Methodology, Writing - original draft, Writing - review and editing; Jean Eberlein, Stefan Baumeister, Hyouk-Bum Kwon, Investigation, Writing - review and editing; Kerstin Wilhelm, Investigation, Writing - review and editing, equal contribution with Toshiya Sugino; Toshiya Sugino, Investigation, Writing - review and editing, equal contribution with Kerstin Wilhelm; Julian Malchow, Annika Schuermann, Hans-Martin Maischein, Investigation; Michael Potente, Wiebke Herzog, Supervision, Funding acquisition, Writing - review and editing; Didier YR Stainier, Conceptualization, Supervision, Funding acquisition, Writing - original draft, Project administration, Writing - review and editing

## Author ORCIDs
Christian SM Helker ⬤ https://orcid.org/0000-0003-0427-5338
Toshiya Sugino ⬤ http://orcid.org/0000-0002-6330-7275
Didier YR Stainier ⬤ https://orcid.org/0000-0002-0382-0026

## Ethics
Animal experimentation: Ethics Statement All zebrafish husbandry was performed under standard conditions, and all experiments were conducted in accordance with institutional (MPG) and national ethical and animal welfare guidelines (Proposal numbers: B2/1017, B2/1041, B2/1218, B2/1138). All procedures conform to the guidelines from Directive 2010/63/EU of the European Parliament on the protection of animals used for scientific purposes.

## Decision letter and Author response
Decision letter https://doi.org/10.7554/eLife.55589.sa1
Author response https://doi.org/10.7554/eLife.55589.sa2

# Additional files

## Supplementary files
• Transparent reporting form

## Data availability
All data generated or analysed during this study are included in the manuscript and supporting files.

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
