## [Decision Letter]

**Acceptance summary:**

The finding that Apelin signaling alters metabolism to drive angiogenesis downstream of the Notch pathway is interesting and novel. The most impressive additions that were made during the revision period to substantiate the author's claims include new panels at 30 hpf showing *apln:EGFP* is more prominent in the tips cells than in the stalk, the increase of filopodia and endothelial cell numbers in apln mutants that show excessive sprouting, suppression of the hyper-sprouting phenotype in dll morphants by *aplnrb* loss-of-function, and *pfkfb3* endothelial cell (EC)-specific overexpression that increases EC metabolic activity rescues sprouting in *aplnrb* mutants. While the conclusions could have been better substantiated by the additional experiments proposed by the reviewers (specifically the experiments that address the link between metabolism and ISV sprouting), the revision is reasonable.

**Decision letter after peer review:**

Thank you for submitting your article "Apelin signaling drives vascular endothelial cells towards a pro-angiogenic state" for consideration by *eLife*. Your article has been reviewed by three peer reviewers, and the evaluation has been overseen by a Reviewing Editor and Marianne Bronner as the Senior Editor. The following individual involved in review of your submission has agreed to reveal their identity: Naoki Mochizuki (Reviewer #2).

The reviewers have discussed the reviews with one another and the Reviewing Editor has drafted this decision to help you prepare a revised submission.

Summary:

The manuscript "Apelin signaling drives vascular endothelial cells towards a pro-angiogenic state" by Helker et al. uses genetic approaches in zebrafish embryos to provide new insights into the regulation of vessel growth by apelin signaling. Specifically, they address expression of Apelin and the Apelin receptor and their functional roles in sprouting intersomitic vessels (ISVs) in the embryonic trunk using new transgenic reporters, existing genetic mutants, and mosaic analyses. Furthermore, they show that Apelin signaling is required for the hypersprouting phenotype resulting from compromised Notch function. A final set of data links Apelin to endothelial cell metabolism.

Essential revisions:

1) The claim that apelin is expressed in tip cells is not well supported by the data provided. It appears that *apln*:*EGFP* staining at 26 hpf is very prominent outside the vasculature (presumably, notochord and spinal cord), which makes it difficult to appreciate the fluorescent signal in tip cells (Figure 1A). The authors should clarify whether *apln* expression is found outside the vasculature and provide clearer images of tip cell expression. In addition, it is important to know whether the sprouts shown on the far left in A' are composed of a single (tip) cell or multiple cells. In case of the latter, it is not obvious that tips are positive whereas stalk cells are not.

2) The cellular mechanism underlying the observed ISV sprouting defect in apelin mutants is not well-defined. Specifically, does apelin primarily regulate filopodia or endothelial cell numbers to alter ISV sprouting? Is it known that tip cell filapodia number correlate with sprout length/distance? If so, do Notch-deficient embryos that show excessive sprouting have increased numbers of tip cell filopodia? Does taking Apelin signaling away from dll morphant embryos decrease filapodia number? It is argued that the overexpression of *apln* does not lead to ectopic sprouting. Nevertheless, the images shown in Figure 2—figure supplement 3 suggest that filopodia numbers, morphology, or extension might be altered. Please clarify.

3) There is an internal inconsistency regarding heightened Apelin expression must be rectified. The authors claim that heightened Apelin expression in Notch-deficient embryos is responsible for the observed hyperspouting phenotype. However, overexpression of Apelin in their hsp70:apelin transgenic line fails to show a hypersprouting phenotype. One possibility is that expression of the Apelin receptor is limiting. Is there a constitutively active apelin receptor that could be used to test whether the Apelin signaling axis is sufficient to promote angiogenesis? Can the hyper-sprouting phenotype observed in Notch-deficient embryos also be suppressed by loss of the Apelin receptor? (Dll4 morpholino into *alpnr-/-* embryos)? Experimental clarification of the model is needed.

4) The link between Apelin and Notch is interesting, but it is unclear whether this effect is mediated by VEGF or increased hypoxia, which were previously linked to the regulation of *apln* expression. The authors should try to clarify the link between Notch and Apelin.

5) The claim that Apelin drives angiogenesis by increasing endothelial cell glycolysis/metabolism is not well supported. Specifically, the connection between *apln* and metabolic alterations has been only explored in cultured cells in vitro and its relevance for the observed phenotypes in zebrafish remains unclear. The authors should validate their key findings in vivo and explore how apelin regulates the metabolic properties of ECs. For example, could 2-DG be used to inhibit glycolysis in zebrafish embryos and ISV sprouting examined? Is glycolysis/metabolism increased in Notch-inhibited vessels that show hypersprouting? Can this phenotype be suppressed by 2-DG? Is there a method to artificially increase metabolism in order to learn whether this is sufficient to promote ISV sprouting? If so, does it rescue sprouting in apelin mutants?

---

## [Author Response]

Essential revisions:1) The claim that apelin is expressed in tip cells is not well supported by the data provided. It appears that apln:EGFP staining at 26 hpf is very prominent outside the vasculature (presumably, notochord and spinal cord), which makes it difficult to appreciate the fluorescent signal in tip cells (Figure 1A). The authors should clarify whether apln expression is found outside the vasculature and provide clearer images of tip cell expression. In addition, it is important to know whether the sprouts shown on the far left in A' are composed of a single (tip) cell or multiple cells. In case of the latter, it is not obvious that tips are positive whereas stalk cells are not.

*apln* expression is enriched in tip cells but is not exclusive to these cells. *apln* expression is also observed in the notochord around 17 hpf (Helker et al., 2015) as well as in the neural tube, and due to the stability of the GFP protein, these tissues are still GFP positive at 26 hpf. Therefore, the EC specific signal is much more visible at 54 hpf when GFP in the notochord and neural tube has mostly been degraded. In Figure 1A’’, the arrowheads point to strong *apln* expression in tip cells, while the arrows point to weak *apln* expression in stalk cells. We added these points in the Results section as well as in the figure legend. In addition, we replaced picture 1A’ with a new one at 30 hpf – a time point when *apln* expression is a bit stronger and easier to visualize.

2) The cellular mechanism underlying the observed ISV sprouting defect in apelin mutants is not well-defined. Specifically, does apelin primarily regulate filopodia or endothelial cell numbers to alter ISV sprouting? Is it known that tip cell filapodia number correlate with sprout length/distance?

Phng et al. reported that the inhibition of filopodia formation by Latrunculin B treatment reduces ISV sprout length (See Figure 5C from Phng et al., 2013), consistent with our hypothesis that the ISV sprouting defects in *apln* mutants are caused by the filopodia defects. However, one cannot exclude the possibility that defects in EC numbers are also contributing to the ISV sprouting defects. These points are now included in the Discussion.

If so, do Notch-deficient embryos that show excessive sprouting have increased numbers of tip cell filopodia?

Several studies have reported that inhibition of Notch signaling increases filopodia numbers (Hellstrom et al., 2007; Leslie et al., 2007; Siekmann et al., 2007; Suchting et al., 2007).

Does taking Apelin signaling away from dll morphant embryos decrease filapodia number?

Phng et al. reported that ectopic sprouting in *dll4* morphants occurs in the absence of filopodia (Phng et al., 2013). Therefore, it is unlikely that the reduction of filopodia observed in *apln/aplnrb* mutants is the reason for the rescue of the hypersprouting phenotype in *apln* or *aplnrb* mutants injected with the *dll4* morpholino.

It is argued that the overexpression of apln does not lead to ectopic sprouting. Nevertheless, the images shown in Figure 2—figure supplement 3 suggest that filopodia numbers, morphology, or extension might be altered. Please clarify.

We thank the reviewers for this valid point. Indeed, overexpression of *apln* leads to defects in lymphatic endothelial cell sprouting. We labeled these defects in Figure 2—figure supplement 3 using arrows and added this additional point to the Results section.

3) There is an internal inconsistency regarding heightened Apelin expression must be rectified. The authors claim that heightened Apelin expression in Notch-deficient embryos is responsible for the observed hyperspouting phenotype. However, overexpression of Apelin in their hsp70:apelin transgenic line fails to show a hypersprouting phenotype. One possibility is that expression of the Apelin receptor is limiting. Is there a constitutively active apelin receptor that could be used to test whether the Apelin signaling axis is sufficient to promote angiogenesis? Can the hyper-sprouting phenotype observed in Notch-deficient embryos also be suppressed by loss of the Apelin receptor? (Dll4 morpholino into alpnr-/- embryos)? Experimental clarification of the model is needed.

We are thankful for these comments and questions. The Apelin receptor is a GPCR and to the best of our knowledge there is no constitutively active form of it available to date. However, we injected the *dll4* morpholino into *aplnrb* mutant embryos, and found that the hypersprouting phenotype observed in Notch-deficient embryos was also suppressed by the loss of the Apelin receptor. These new data can be found in supplemental Figure 4—figure supplement 1.

4) The link between Apelin and Notch is interesting, but it is unclear whether this effect is mediated by VEGF or increased hypoxia, which were previously linked to the regulation of apln expression. The authors should try to clarify the link between Notch and Apelin.

The transcriptional regulation of Apelin Signaling components by Notch signaling is very interesting. However, it is very unlikely that *apln* is a direct Notch target gene since activation of Notch signaling leads to a downregulation of *APLN* expression. Therefore, we speculate that *apln* expression is regulated by downstream Notch target genes such as those encoding members of the HES/HEY Transcription factor family, which are known to act as transcriptional repressors. This point has now been included in the Discussion. Thus, further work is required to obtain a detailed view of the regulation of Apelin signaling components by Notch signaling.

5) The claim that Apelin drives angiogenesis by increasing endothelial cell glycolysis/metabolism is not well supported. Specifically, the connection between apln and metabolic alterations has been only explored in cultured cells in vitro and its relevance for the observed phenotypes in zebrafish remains unclear. The authors should validate their key findings in vivo and explore how apelin regulates the metabolic properties of ECs. For example, could 2-DG be used to inhibit glycolysis in zebrafish embryos and ISV sprouting examined? Is glycolysis/metabolism increased in Notch-inhibited vessels that show hypersprouting? Can this phenotype be suppressed by 2-DG? Is there a method to artificially increase metabolism in order to learn whether this is sufficient to promote ISV sprouting? If so, does it rescue sprouting in apelin mutants?

It has been shown that stimulation of Apelin signaling induces the phosphorylation, and therefore nuclear exclusion, of FOXO1 (Figure 3B from Hwangbo et al., 2017). Furthermore, expression of a constitutively active FOXO1, which remains in the nucleus, has been shown to reduce MYC protein levels (Figure 3K from Wilhelm et al., 2016), and reduce metabolic activity (Figure 3E from Wilhelm et al., 2016). In agreement with these in vitro data, deleting *Foxo1* in mouse ECs enhances metabolic activity and promotes EC sprouting in vivo (Figure 1B/C from Wilhelm et al., 2016).

Since 2-DG treatments not only affect ECs but the whole organism, we overexpressed *pfkfb3* specifically in ECs in the *aplnrb* mutant background. These experiments show that enhancing the metabolic activity of ECs can rescue the sprouting phenotype of *aplnrb* mutant embryos. These new data can be found in Figure 5.